# Study on the Degradation Effect of Carbonaceous Shale under the Coupling Effect of Chemical Erosion and High Temperature

**DOI:** 10.3390/ma17030701

**Published:** 2024-02-01

**Authors:** Guangwei Xiong, Qiunan Chen, Yongchao He, Zhenghong Chen, Xiaocheng Huang, Yunpeng Xie

**Affiliations:** 1College of Civil Engineering, Hunan University of Science and Technology, Xiangtan 411201, China; yzpx775474208@gmail.com (G.X.); chenzhenghong92@126.com (Z.C.); hxc@hnust.edu.cn (X.H.); 2Hunan Province Key Laboratory of Geotechnical Engineering Stability Control and Health Monitoring, Hunan University of Science and Technology, Xiangtan 411201, China; 3College of Resource & Environment and Safety Engineering, Hunan University of Science and Technology, Xiangtan 411201, China; nmmwm1@163.com; 4Research Center of Geotechnical and Structural Engineering, Shandong University, Jinan 250061, China

**Keywords:** carbonaceous slate, chemical erosion, triaxial compression, discrete element simulation

## Abstract

The southwest region of China has abundant groundwater and high-temperature geothermal energy. Carbonaceous shale, as one of the typical surrounding rocks in this region, often suffers from deterioration effects due to the coupled action of groundwater chemical erosion and high temperature, which affects the long-term stability of tunnel engineering. In order to investigate the deterioration effects of carbonaceous shale under the coupled action of chemical erosion and high temperature, carbonaceous shale from a tunnel of Lixiang Railway in Yunnan Province was taken as the research object. The microstructure and mineral composition of the samples before and after chemical erosion were obtained with a scanning electron microscope-energy dispersive spectrometer and an X-ray diffraction test. Then, triaxial compression tests were conducted on the samples under different time points and different temperature effects of chemical erosion, and the stress–strain curves and the deterioration laws under a single factor were obtained. An improved numerical simulation method based on the parallel bond model was developed, which can account for the coupled effects of chemical erosion and high temperature on the rock. By simulating the triaxial compression test of carbonaceous shale, the deterioration law of carbonaceous shale under the coupled action was discussed. The results show that chemical erosion has a significant deterioration effect on the triaxial compressive strength of carbonaceous shale, and the degree of deterioration is related to the erosion time. In the first 30 days of erosion, the triaxial compressive strength of carbonaceous shale decreased by 11.38%, which was the largest deterioration range. With the increase in erosion time, the deterioration rate gradually decreased; temperature had a significant threshold effect on the strength of carbonaceous shale, and a clear turning point appeared at about 200 °C. By simulating the deterioration effects of carbonaceous shale under the coupled action of chemical erosion and high temperature, it was found that the longer the duration of chemical erosion, the stronger the temperature sensitivity of carbonaceous shale, and the more serious the loss of compressive strength during the heating process. When the temperature was low, the strength of carbonaceous shale changed little, and some samples even showed an increase in strength; when the temperature was high, the strength of carbonaceous shale decreased significantly, showing deterioration characteristics. The numerical simulation method was compared and verified with the indoor test results, and it was found that the numerical calculation had a good agreement with the test results.

## 1. Introduction

The mechanical properties of deep rock are the basis for developing deep earth resources and exploring the laws of deep earth science [1,2]. The Lixiang Railway, as an important part of the Yunnan–Tibet Railway in China, has many deep-buried tunnels, with a maximum burial depth of 687 m. The Lixiang Railway is located at the southeastern edge of the Qinghai–Tibet Plateau, in the middle section of the Hengduan Mountains, and belongs to the plateau river erosion valley landform with prominent geological disasters along the line [3]. Carbonaceous shale, as a typical surrounding rock in southwestern China, often appears in the tunnels along the line. The physical and mechanical properties of carbonaceous shale are affected by different degrees under the combined action of temperature and groundwater chemical erosion, which poses a great challenge to the long-term stability of tunnel engineering [4,5,6]. Therefore, the research on the deterioration effect of carbonaceous shale under chemical erosion and high temperature is of great practical significance for China to explore the laws of deep earth science and tunnel engineering design and construction in the future [7,8,9]. 

The research on carbonaceous shale can be mainly divided into experimental research, numerical simulation, and theoretical derivation [10]. In terms of experimental research, Li et al. [11] carried out Brazilian splitting tests on carbonaceous shale after instant drying and static weathering, and analyzed the tensile mechanical properties of carbonaceous shale with different bedding angles under the two test methods. Xie et al. [12] used a scanning electron microscope-energy dispersive spectrometer (SEM-EDS) and an X-ray diffraction (XRD) to characterize the microstructure and mineral composition of carbonaceous shale, carried out uniaxial compression tests on carbonaceous shale with different water contents, analyzed its stress–strain relationship, deformation characteristics, and failure mode, and discussed the influence mechanism of water content on its mechanical properties. In terms of numerical simulation, Li et al. [13] established a transversely isotropic elastic–plastic constitutive model of carbonaceous shale, and obtained the material parameters of carbonaceous shale with different bedding angles by triaxial compression tests. Then, the mechanical behavior of carbonaceous shale under different confining pressures was numerically simulated by using the subroutine user-defined material mechanical behavior (UMAT) in ABAQUS. In terms of theoretical derivation, Wang et al. [14] established a transversely isotropic creep constitutive model of rock, verified it with experimental data, and carried out creep parameter inversion research. In summary, in the research on the mechanical properties of carbonaceous shale, the influence of environment on rock is considered less; some studies consider the effects of weathering and water, while in engineering practice, the factors that affect rock in the environment are more complex. 

Regarding the mechanical properties of rock under environmental influence, Haneef et al. [15] used SEM and XRD tests to study the corrosion of different rocks under acidic solution erosion, and concluded that acidic solution would cause erosion on rock. Feucht et al. [16] analyzed the variation of shear, friction, and other strength characteristics of sandstone under different water chemical solutions by related mechanical tests. Xie et al. [17] carried out indoor graded loading creep tests, constructed a hardening damage creep model that can respond to the accelerated creep stage, derived the three-dimensional creep differential equation under different geostress states, developed Fast Lagrangian Analysis of Continua (FLAC) to establish a new model, and verified and applied the new model. Wang et al. [18] conducted various experiments and tests to reveal the decomposition law, microstructural change, and fractal characteristics of carbonaceous shale in high-latitude cold regions under the wetting–drying cycle and freeze–thaw action, as well as the main factors affecting its decomposition. Li et al. [19] simulated the heat transfer and thermal expansion in rock by a particle flow program, calibrated the mesoscopic parameters according to the different thermodynamic parameters of mineral components in rock, and explored the influence of crack opening distribution under different shape parameters on rock thermal conduction and the thermal cracking process. At present, the exploration of the multi-factor coupling mechanical properties of rock is relatively in-depth, but the research on carbonaceous shale mainly focuses on the single-factor influence; therefore, it is necessary to explore the influence of multi-factor coupling on carbonaceous shale more deeply [20,21]. 

To explore the deterioration effect of carbonaceous shale under temperature and chemical erosion, this experiment carried out indoor tests on the triaxial compressive strength of carbonaceous shale specimens at different soaking time points, obtained the mineral composition of carbonaceous shale at each time point by SEM-EDS and XRD tests, established a parallel bond model, and compared and verified it with the test results. Through the verified numerical model, the mesoscopic damage and macroscopic deterioration effect of carbonaceous shale under the coupled action of high temperature and chemical erosion were further explored, which has certain reference significance for the research on the mechanical properties of other rock masses in deep high-temperature environments, and also provides some reference for the numerical calculation of engineering.

## 2. Sampling and Test Procedures

### 2.1. Test Sampling

This experiment selected rock samples from a certain tunnel of the Lixiang Railway in Yunnan Province, as shown in Figure 1. The collected carbonaceous shale rock body is gray, relatively dense in structure, and has a significant bedding plane. Standard specimens with a diameter of 50 ± 0.5 mm and a height of 100 ± 0.5 mm were prepared, with an average density of 2.608 g/cm^3^. 

After preparing the specimens, the specimens were grouped, as shown in Figure 2. For the carbonaceous shale test under chemical erosion conditions, there were no fewer than 3 specimens in each group under different groups and 5 specimens under temperature conditions, including specimens to prevent test operation errors, totaling 20 specimens. Different time points were marked with four different colors: green, yellow, pink, and blue. Uniaxial compression tests were carried out on the specimens, and the compressive strength of the specimens was 73.43 MPa, the elastic modulus was about 36.02 GPa, and the Poisson’s ratio was 0.26.

### 2.2. Test Preparation

The following paragraph describes the chemical erosion effects of SO_4_^2−^, Cl^−^, and H^−^ ions in the groundwater of the tunnel area on carbonaceous shale. To simulate the chemical erosion caused by groundwater on carbonaceous shale, dilute hydrochloric acid, NaCl, and Na_2_SO_4_ solutions were used [3]. Table 1 shows the solution preparation details. The specimens to be eroded were grouped and labeled with waterproof tags, and then immersed in an acidic chemical solution for 90 days. They were taken out and dried at 30, 60, and 90 days, respectively [22,23]. The soaking box, as shown in Figure 3, includes a temperature control system and a circulating water pump. The temperature control system ensures that the temperature does not affect the samples during the erosion process, and the circulating water pump ensures that each sample is exposed to similar erosion conditions. In actual tunnel engineering, the maximum horizontal ground stress measured in deep holes can reach 38.84 MPa, and the highest groundwater temperature can reach 39 °C. The soaking temperature was set to 40 °C. The pH value of the solution was maintained at around 5. To ensure that the concentration of the solution was the same during the soaking process, the pH value of the solution was measured twice a day, and the solution was supplemented according to the volatilization of the solution. 

The loading equipment used in the experiment was a TAW-2000 (Changchun Chaoyang Test Instrument Company, Changchun, China) rock creep triaxial test machine, with a maximum axial pressure of 2000 kN and a maximum confining pressure of 100 Mpa, as shown in Figure 4. It can monitor the deformation: axial 0–5 mm, radial 0–3 mm, measurement resolution 0.0001 mm, measurement accuracy ± 1%, deformation speed control range 0.01 mm/min–50 mm/min. The instrument was equipped with a heating system with a maximum loading temperature of 150 °C.

## 3. Laboratory Tests

### 3.1. Microscopic Changes of Carbonaceous Shale under Chemical Erosion

In order to observe the changes at the microscopic level of the carbonaceous slate, tests were carried out using a Gemini Sigma 300 (Carl Zeiss Company, Oberkochen, Germany) electron microscope and a Smart EDX kit. After soaking in different chemical solutions for different periods, the microstructure of carbonaceous shale was eroded and damaged to different extents. Figure 5 shows the SEM images of the surface microstructure of carbonaceous shale. As can be seen from Figure 5a, the uneroded carbonaceous shale showed a distinct layered structure, with step-like distribution between the layers, and the step surface was relatively smooth and dense; as shown in Figure 5b, after 90 days of erosion, the layered structure of carbonaceous shale almost disappeared and was replaced by a large number of pits, and some microcracks were covered or filled by loose particles. These results indicate that the chemical solution had a significant effect on the microstructure of carbonaceous shale, causing changes in its surface morphology and pore structure.

Figure 6 shows some changes of elements in carbonaceous slate before and after 90 days of immersion. It can be seen that the proportion of Al element decreases significantly, while the proportions of Si and C elements change slightly. The two main mineral components of carbonaceous slate in this region are muscovite (KAl_2_ [AlSi_3_OH_10_] (OH)_2_) and albite (Na[AlSi_3_O_8_]), which both contain Al ions. A qualitative analysis can be performed, and these two minerals undergo hydrolysis reactions in a weak acid environment, which affects the mechanical properties of carbonaceous slate.

In order to quantitatively analyze the mineral composition changes in carbonaceous slate under chemical erosion, XRD tests were carried out on carbonaceous slate with different erosion times; the test instrument used was a D8 Advance-type X-ray diffractometer (Bruker Corporation, Karlsruhe, Germany), which obtained the mineral composition change data as shown in Table 2. It can be found that: with the increase of chemical erosion time, due to the occurrence of chemical reactions, the proportion of reactive minerals, such as muscovite and albite in carbonaceous shale, decreased significantly, while the proportion of non-reactive or difficult-to-react minerals, such as quartz and graphite, increased significantly. This change in mineral composition will lead to the change in physical properties such as density, porosity, and mechanical characteristics such as elastic modulus, Poisson’s ratio, and compressive strength of carbonaceous shale.

### 3.2. Strength Deterioration of Carbonaceous Shale under Chemical Erosion

To control the variables, the temperature was controlled at 40 °C and the confining pressure was controlled at 40 Mpa during the test. This was close to the temperature and confining pressure measured in the engineering project, and could provide reference for the physical and mechanical properties of carbonaceous shale in practical engineering while exploring the laws [24]. The changes in triaxial peak strength at different time steps in each group are shown in Figure 7. 

Figure 7a shows the stress–strain curves of carbonaceous shale with different chemical erosion durations in triaxial compression tests. It can be seen from the figure that the peak deviatoric stress of carbonaceous shale decreases continuously with the increase of chemical erosion time, and the brittleness of the uneroded sample is significantly higher than that of the other three groups of samples with different erosion durations. This indicates that chemical erosion will reduce the strength and brittleness of carbonaceous shale and increase its plastic deformation capacity.

Figure 7b shows the peak deviatoric stress of carbonaceous shale with different chemical erosion durations, combined with the values of the dilatancy onset deviatoric stress in Figure 7a. It can be seen that, according to different chemical erosion times, the peak deviatoric stress of the uneroded sample was 183.23 MPa, and the dilatancy onset deviatoric stress was 153.71 MPa; the peak deviatoric stress of the sample eroded for 30 days was 162.37 MPa, and the dilatancy onset deviatoric stress was 149.81 MPa; the peak deviatoric stress of the sample eroded for 60 days was 144.72 MPa, and the dilatancy onset deviatoric stress was 117.33 MPa; the peak deviatoric stress of the sample eroded for 90 days was 137.34 MPa, and the dilatancy onset deviatoric stress was 102.46 MPa. The triaxial compressive strength softening degrees were 11.38%, 21.02%, and 25.05%, respectively. It was found that—consistent with the trend of the ultrasonic longitudinal wave velocity of the rock samples—the rock samples softened continuously with the increase of chemical erosion time, but the softening rate decreased significantly. 

During the test, the concentration and pH value of the chemical solution were measured twice a day and supplemented according to the evaporation of the solution, to ensure that the effective components of the chemical solution would not decrease due to the long-term erosion. Therefore, the possibility that the decrease of the effective components of the chemical solution caused the decrease of the triaxial compressive strength deterioration rate of carbonaceous shale can be ruled out. In this study, it is considered that in the process of chemical erosion, the reactive minerals inside the rock sample have their erosion limit. After reaching this limit, the development of microcracks and pores inside the rock sample will slow down, leading to the slowdown of the strength softening rate of the sample, but the strength softening process will not stop completely.

### 3.3. The Effect of High Temperature on the Strength of Carbonaceous Shale

To explore the variation law of compressive strength of carbonaceous shale in deep and high-temperature environments, uneroded samples were selected for triaxial compression tests under different temperature conditions. Due to the limitation of test equipment, the temperature range was chosen from 40 °C to 120 °C, with 20 °C as a node, and five groups of tests were conducted.

Figure 8 shows the strength variation curves and elastic modulus variation curves of carbonaceous shale in triaxial compression tests under different temperatures. It can be seen in Figure 8 that with the increase in temperature, the peak deviatoric stress of carbonaceous shale decreased by 10.61%, which was a small change, and the elastic modulus decreased by 31.02% compared with the condition of 40 °C and 120 °C, which was more obvious. This indicates that the increase in temperature will reduce the strength and brittleness of carbonaceous shale, and increase its plastic deformation capacity, but the influence on the strength is small below 120 °C.

### 3.4. Failure Characteristics Analysis of Carbonaceous Shale

In the triaxial compression test, the sample failure was mainly characterized by vertical penetrating cracks, accompanied by sliding displacement cracks with the same dip angle as the bedding, as shown in Figure 9. The crack formation was significantly influenced by the bedding dip angle and mainly showed oblique penetrating cracks, indicating that the layered distribution had an important role in the crack formation of the sample. During the failure of carbonaceous shale, cracks were usually initiated at both loading ends, and under the influence of confining pressure, they were more inclined to occur in the middle of the loading end; however, due to the bedding dip angle, in addition to the initial cracks in the middle of the loading end, the cracks in the sample often extended along the bedding. Under the action of chemical solution erosion, the cracks along the bedding became gradually obvious, indicating that chemical erosion had an accelerating erosion effect on the interlayer sliding. On the front and back of the sample, surface cracks penetrated at the end and caused block detachment, indicating that there was a stress concentration effect at the end of the sample.

## 4. Numerical Simulation

The lab tests of carbonaceous shale have some problems: safety, observation means, economic cost, etc. It is hard to study the deterioration effect above 150 °C and the mesoscopic damage of carbonaceous shale under high temperature is not directly observable. In this study, a numerical simulation method based on the parallel bond model is improved, which can simultaneously consider the coupled effects of chemical erosion and high temperatures on rocks. This method can explore the macroscopic deterioration effect and observe the mesoscopic damage of the sample through the discrete crack network.

### 4.1. Parallel Bond Model

The triaxial compression tests were simulated using particle flow code (PFC 5.0) software, and the specimens were modeled based on a linear parallel bond model. The advantage of using PFC software is that the model is not limited by the amount of deformation, which can effectively simulate discontinuous phenomena, such as cracking and separation of the medium, and can react to the mechanism, process, and results [25]. The linear parallel bond model provides the behavior of two interfaces: an infinitesimal linear elastic (no tension) and frictional interface that bears a force, and a finite-sized linear elastic and bonded interface that bears a force and a moment. The first interface is equivalent to the linear model; it does not prevent relative rotation, and controls slip by applying a Coulomb limit to the shear force. The second interface is called a parallel bond, which acts parallel to the first interface when it is bonded. When the second interface is bonded, it prevents relative rotation, and its behavior is linear elastic until it exceeds the strength limit and the bond breaks, causing it to fail. When the second interface is not bonded, it does not bear any load. When the stress exceeds the corresponding bond strength, the parallel bond contact breaks, and the transmitted force, moment, and stiffness disappear, degrading to the linear model.

The calculation formulas of the force and moment transmitted by the parallel bond between particles under external load are [26]: $$\begin{array}{c} F_{i}^{n}={\bar{F}}_{i}^{n}+2\lambda R_{i}(\bar{E}/L)\Delta \delta_{i}^{n}\\ F_{i}^{s}={\bar{F}}_{i}^{s}+2\lambda R_{i}(\bar{E}/\bar{k}L)\Delta \delta_{i}^{s}\\ M_{i}^{n}={\bar{M}}_{i}^{n}+\frac{2}{3}\lambda^{3}R_{i}^{3}(\bar{E}/L)\Delta \theta_{i}^{n} \end{array}$$

Stress on the parallel bond:$$\begin{array}{c} \sigma_{i}=-\frac{F_{i}^{n}}{2\lambda R_{i}}+\frac{3|M_{i}^{n}|}{2\lambda^{2}R_{i}^{2}}\\ \;\;\;\;\tau_{i}=\frac{|F_{i}^{s}|}{2\lambda R_{i}} \end{array}$$

Failure condition of the parallel bond:$$\begin{array}{c} \sigma_{i}>\sigma_{c}\\ \tau_{i}>\tau_{c}=c-\sigma_{i}\tan\;\phi \end{array}$$
where ${\bar{F}}_{i}^{n}$, ${\bar{F}}_{i}^{s}$, ${\bar{M}}_{i}^{n}$ are the normal contact force, shear contact force, and normal moment of the parallel bond at the i-th contact before the time step, $F_{i}^{n}$, $F_{i}^{s}$, $M_{i}^{n}$ are the normal contact force, shear contact force, and normal moment of the parallel bond at the i-th contact, $\sigma_{i}$ is the parallel bond radius multiplier, $R_{i}$ is the parallel bond radius of the i-th contact, $\bar{E}$, $\bar{k}$ are the parallel bond modulus and stiffness ratio, $\tau_{i}$ are the normal stress and shear stress of the parallel bond at the i-th contact, $\sigma_{c}$, $\tau_{c}$ are the tensile strength and shear strength of the parallel bond, c, ϕ are the cohesion and internal friction angle of the parallel bond.

### 4.2. Numerical Model Establishment

To establish the parallel bond model, we need to calibrate its mesoscopic parameters. The mesoscopic parameters include the shape, size, distribution, density, stiffness, etc., of the particles, which directly affect the macroscopic mechanical properties of the model [27,28]. Based on the macroscopic mechanical parameters of the uniaxial compression test, we used the inverse analysis method by adjusting the values of the mesoscopic parameters, to make the stress–strain curve of the model as close as possible to the test curve [29]. Table 3 shows the calibrated mesoscopic parameters and their value ranges.

Since chemical erosion will cause the weakening of the bond between the particles of carbonaceous shale, the key issue to be considered in establishing the parallel bond model is how to better simulate the macroscopic mechanical parameter changes with the mesoscopic parameter changes. Under the action of chemical erosion, the cementing material dissolves and migrates, resulting in the loss of cementing material and the expansion of micro-defects. It can be assumed that the penetration of chemical erosion is a uniform and constant process, that is, the amount of cementing material reduction can reflect the degree of chemical erosion. Based on this assumption, this study introduces the amount of cementing material reduction after weakening as the chemical erosion factor in the model.

PFC software itself provides a thermo-mechanical coupling module, but it cannot fully simulate all the effects of temperature on rock [30,31]. The PFC thermo-mechanical coupling module is divided into two parts: one is heat transfer, and the other is thermal expansion and contraction. The former is the influence of rock properties on the thermal field, and the latter is the influence of the thermal field on the rock. The thermodynamic parameters of rock in PFC are divided into three parts: the thermal expansion coefficient (thexp) reflects the thermal expansion and contraction of rock due to temperature changes, while the thermal resistance (thres) and specific heat capacity (sheat) reflect the heat transfer between rocks.

As the temperature continues to rise, intergranular cracks will occur due to the different thermal expansion coefficients of different mineral components, and then as the temperature gradually increases, the number of microcracks begins to increase sharply [32]. Since some minerals with low content are not considered in the modeling, the thermodynamic parameters of the main minerals contained in the sample are shown in Table 4 [33]. When performing heat treatment, the mineral components need to be grouped and the thermal expansion coefficient and heat transfer coefficient of each mineral need to be set separately. In the simulation process, the mineral components with low content are ignored, and only the effects of the four main minerals are considered. The minerals are distributed proportionally inside the model, and their composition is similar to the results obtained by scanning electron microscopy and XRD tests. The final model of the carbonaceous shale sample generated after the modeling is shown in Figure 10.

### 4.3. Simulation Results Verification and Comparison

According to the results of the laboratory triaxial compression test of the carbonaceous shale samples, the stress–strain results under the same model size and loading mode are compared. As shown in Figure 11, the simulation effect is the best under the uneroded condition, with a peak deviatoric stress error of 0.25% and an elastic modulus error of 0.54%; the simulation effect is the worst when eroded for 90 days, with a peak deviatoric stress error of 4.71% and an elastic modulus error of 9.34%. The reason for the increase of the error may be that the change in the microstructure of the sample caused by chemical erosion cannot be well simulated in addition to the reduction of the cementing material. After comparison, the simulation results are consistent with the test results in terms of the variation trend, and the simulation effects of the elastic modulus and the peak deviatoric stress are good, but the simulation effect of the Poisson’s ratio is general; this may be because the carbonaceous shale often produces failure and displacement along the bedding plane during the triaxial compression process, which has a great influence on the radial deformation.

The simulation results of carbonaceous shale deterioration under high temperature were further compared. As shown in Figure 12, under the temperature condition of 80 °C, the simulation effect of carbonaceous shale deterioration under high temperature was good, with an elastic modulus error of 4.41% and a peak deviatoric stress error of 1.33%. The maximum error of elastic modulus between the model and the test was 22.54%, and the maximum error of peak deviatoric stress was 6.4%. The reason for the large error of elastic modulus may be that in the numerical simulation process, the thermal expansion of particles did not reach the level of producing intergranular cracks, and the elastic modulus did not change significantly. Generally speaking, the simulation results of the elastic modulus curve and the peak deviatoric stress curve were consistent with the test results.

### 4.4. Mesoscopic Damage of Rock under the Coupled Action

To show the microcracks formed after the contact fracture, this study uses the DFN discrete fracture network function provided by the PFC software, and generates cracks between the particles according to the relationship between the particle stress and the set parallel bond tensile strength or shear strength. The principle is that when the particle stress exceeds the set parallel bond tensile strength or shear strength, the parallel bond between the particles breaks and forms cracks. As shown in Figure 13, the microcracks generated after the sample is heat-treated are shown in blue. Through simulation, the number and distribution of microcracks can be intuitively observed, which is conducive to analyzing the law of temperature influence on the sample.

Table 5 shows the number of microcracks generated by rock samples under different chemical erosion durations and temperatures. It can be seen from the table that when the temperature reaches 100 °C, almost no microcracks are produced in the rock samples. At this time, the samples only undergo thermal expansion, and the thermal stress has not exceeded the inter-particle bond strength set by the model, so it does not cause inter-particle fracture. When the temperature reaches 200 °C, some microcracks occur between some particles, but the number of cracks is small and has little effect on the sample. After the temperature exceeds 300 °C, the number of microcracks increases rapidly, indicating that the thermal stress between some minerals begins to exceed the bond strength. The effect of chemical erosion duration on microcracks conforms to the law found in the laboratory tests. The effect of chemical erosion is most obvious in the first 30 days, because at this time the mineral composition on the surface and inside of the rock has changed greatly, resulting in a reduction of the inter-particle bond strength. After exceeding 300 °C, the erosion effect gradually decreases, and the erosion for 60 days and 90 days has almost no effect on the number of microcracks. This is because the rock erosion has reached a certain equilibrium state, the chemical reaction rate decreases, and the temperature damage to the rock has already dominated.

As shown in Figure 14, by simulating the triaxial compression test of carbonaceous shale under the combined influence of temperature and chemical erosion, it was found that mesoscopic damage had a significant impact on the macroscopic compressive strength. Temperature had a significant threshold effect on the strength of carbonaceous shale. When the simulated temperature reached 200 °C, the compressive strength loss of the uneroded specimens and the specimens eroded for 30 days, 60 days, and 90 days was −1.58%, −0.27%, 0.33%, and 0.39%, respectively, and some specimens even showed an increase in strength. After the temperature exceeded 200 °C, the triaxial compressive strength decreased. The peak triaxial compressive strength of the uneroded specimens at 400 °C was 97.06 Mpa, which was 41.35% lower than the compressive strength of the specimens at 40 °C. The peak strength of the specimens eroded for 30 days, 60 days, and 90 days at 400 °C was 97.06 Mpa, 83.46 Mpa, and 78.23 Mpa, respectively, and the compressive strength loss was 41.50%, 42.77%, and 43.14%, respectively. Obviously, chemical erosion had an effect on the triaxial compressive strength of carbonaceous shale under high temperature. With the increase of chemical erosion time, the temperature sensitivity of carbonaceous shale became stronger, and the compressive strength loss of rock during the heating process became more serious.

## 5. Conclusions

The long-term stability of tunnels is largely influenced by the strength of surrounding rocks, and the investigation of the deterioration effects of rocks under complex conditions has some reference significance for the long-term strength change of surrounding rocks. In this study, a chemical erosion environment similar to that in tunnel engineering in the southwest region was established, and micro-level tests of SEM-EDS and XRD were conducted on carbonaceous slate under the influence of chemical erosion. And through the macro-level triaxial compression test, the deterioration law of carbonaceous slate under the single-factor influence of chemical erosion or temperature was explored. An improved numerical simulation method based on the parallel bond model was developed, which can account for the coupled effects of chemical erosion and high temperature on the rock, and the accuracy of the model was verified by comparing the stress–strain curves of the test. The conclusions are as follows: The peak deviatoric stress of the uneroded and eroded specimens for 30 days, 60 days, and 90 days was 183.23 MPa, 162.37 MPa, 144.72 Mpa, and 137.34 MPa, respectively. The triaxial compressive strength softening degrees were 11.38%, 9.64%, and 4.03%, respectively. This indicates that with the passage of time, the influence of chemical erosion on carbonaceous shale gradually weakened, the compressive strength of the specimens decreased faster in the early stage of erosion, and then the decrease rate gradually decreased.

The failure characteristics of carbonaceous shale indicate that under the triaxial compression state, the fracture of rock samples is mainly vertical through cracks, accompanied by slip displacement cracks along the bedding of rock samples. This fracture form is related to the stress state and bedding direction of rock samples. When the confining pressure is high, the principal stress direction of rock samples is close to parallel to the bedding direction, resulting in shear slip along the bedding. Compared with the uniaxial compression test results, the tensile splitting failure along the axis is obviously suppressed by the confining pressure, while the shear slip failure along the bedding is obviously enhanced. This shows that the confining pressure has a significant influence on the fracture form and strength of rock.

An improved numerical simulation method better simulates the coupled effect of chemical erosion and temperature on the deterioration of carbonaceous slate by considering the changes of bond strength and mineral composition at the mesoscopic level. This method makes the deterioration of carbonaceous slate under the coupled effect closer to the actual working conditions, and lays a foundation for predicting the long-term strength of surrounding rock. This method is not limited to carbonaceous slate and has universality. It can provide a reference for the study of rock deterioration under multiple factors.

By simulating the triaxial compression test of carbonaceous slate under the coupled action of chemical erosion and high temperature, it was found that temperature had a significant threshold effect on the strength of carbonaceous slate, and a clear turning point appeared at about 200 °C. When the simulated temperature increased from 40 °C to 200 °C, the triaxial compressive strength of the four samples with different erosion durations changed slightly. The uneroded sample and the sample eroded for 30 days increased by 1.58% and 0.27%, respectively, while the samples eroded for 60 days and 90 days decreased by 0.33% and 0.39%, respectively. The compressive strength losses of the uneroded sample and the samples eroded for 30 days, 60 days, and 90 days at 400 °C were 41.35%, 41.50%, 42.77%, and 43.14%, respectively. From the deterioration degree of the four samples with different erosion durations, it can be seen that the longer the duration of chemical erosion, the stronger the temperature sensitivity of carbonaceous slate, and the more serious the loss of compressive strength during the heating process.

## Figures and Tables

**Figure 1 materials-17-00701-f001:**
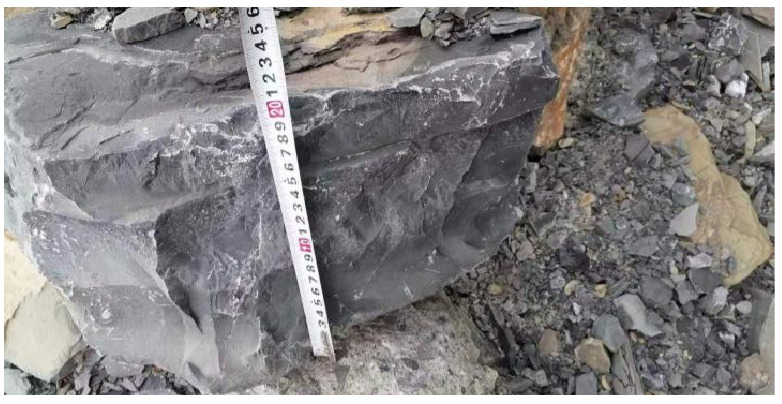
The carbonaceous slate rock sample.

**Figure 2 materials-17-00701-f002:**
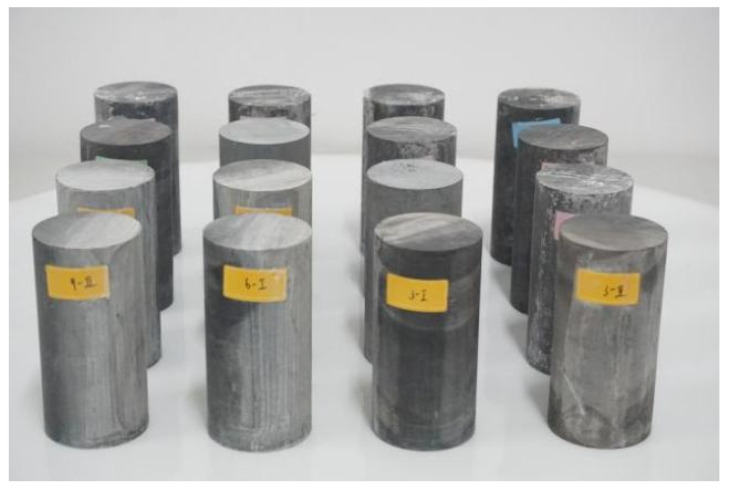
Partial carbonaceous slate specimen.

**Figure 3 materials-17-00701-f003:**
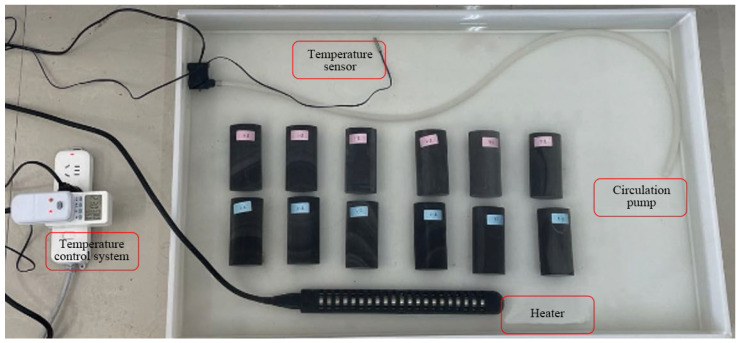
The chemical erosion soaking box.

**Figure 4 materials-17-00701-f004:**
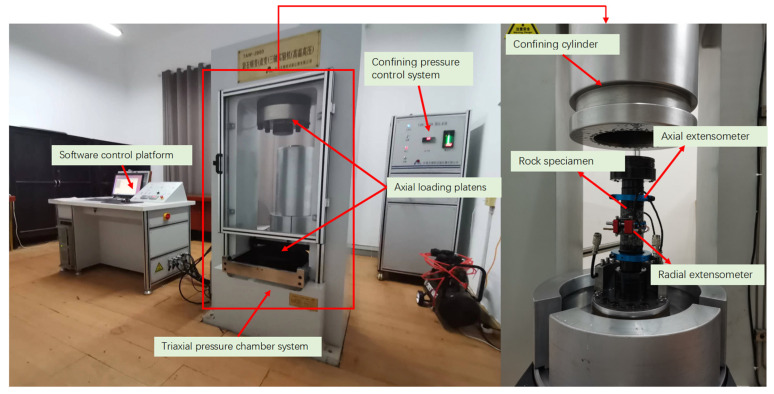
TAW-2000 rock creep triaxial testing machine.

**Figure 5 materials-17-00701-f005:**
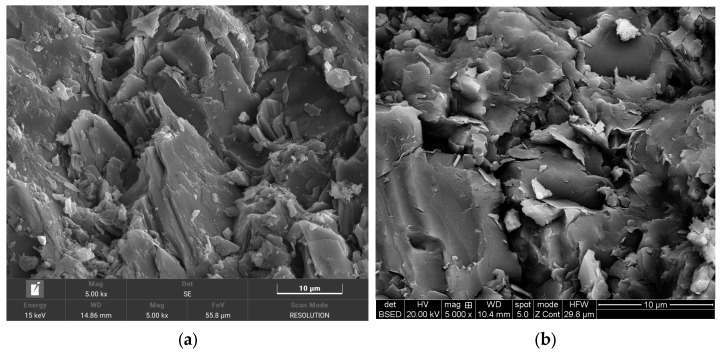
SEM results before and after chemical erosion. (**a**) Uneroded specimen. (**b**) Erosion of 90-day specimen.

**Figure 6 materials-17-00701-f006:**
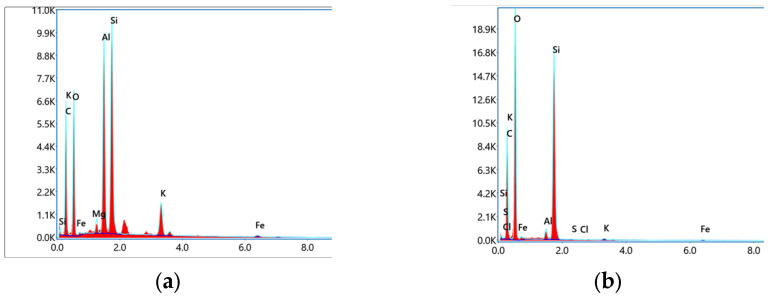
Selected elemental energy spectra before and after chemical erosion. (**a**) Uneroded specimen. (**b**) Erosion of 90-day specimen.

**Figure 7 materials-17-00701-f007:**
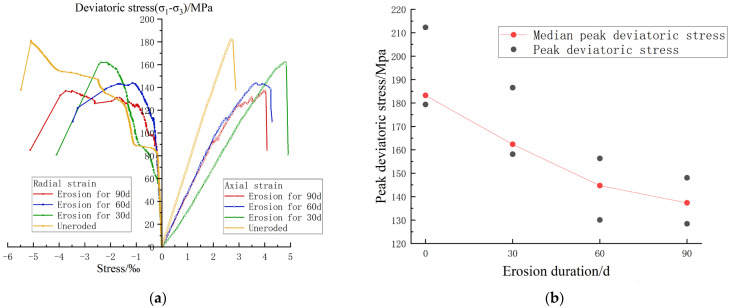
The influence of chemical erosion on compressive strength. (**a**) Stress–strain curve for median specimen. (**b**) Peak deviatoric stress change of specimens under chemical erosion.

**Figure 8 materials-17-00701-f008:**
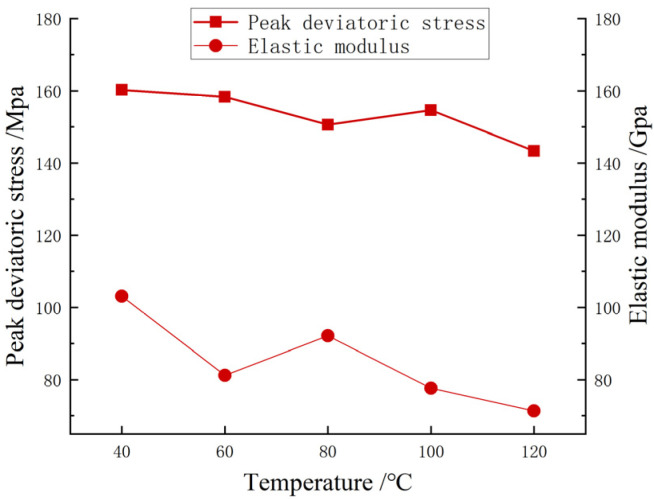
The influence of chemical erosion on compressive strength.

**Figure 9 materials-17-00701-f009:**
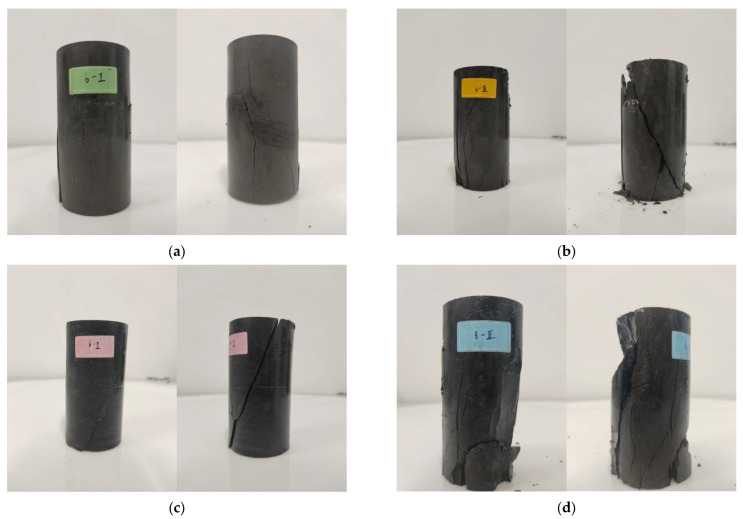
Typical failure characteristics of specimens. (**a**) Uneroded specimen. (**b**) Erosion of 30-day specimen. (**c**) Erosion of 60-day specimen. (**d**) Erosion of 90-day specimen.

**Figure 10 materials-17-00701-f010:**
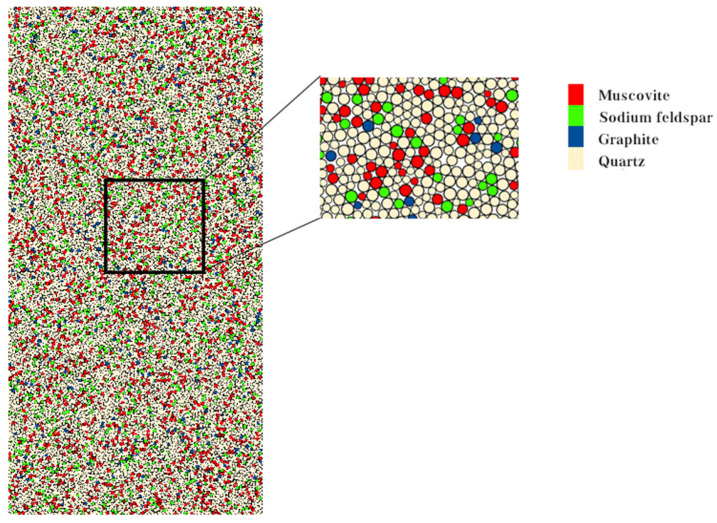
Particle flow model and mineral composition schematic.

**Figure 11 materials-17-00701-f011:**
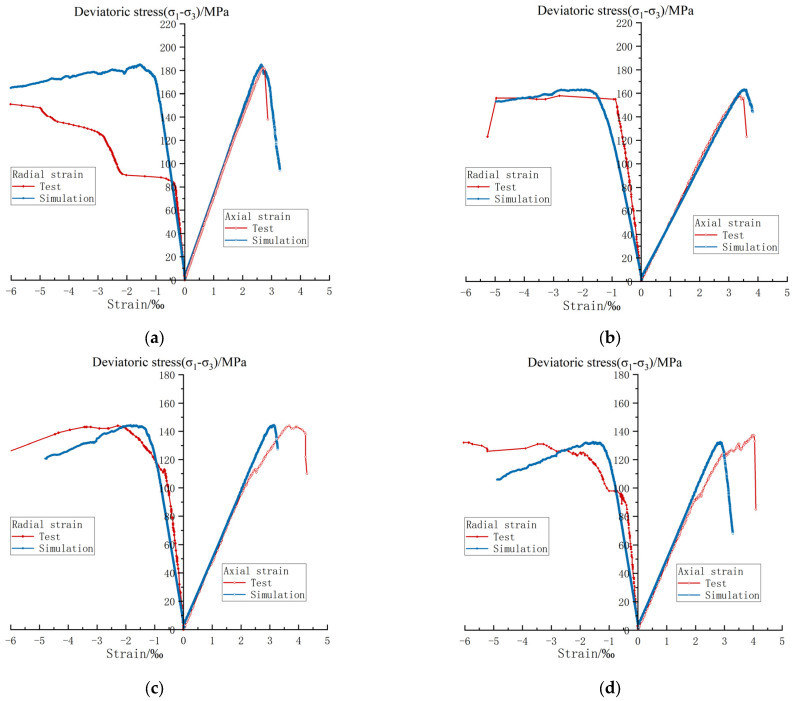
Comparison of stress–strain curves between indoor tests and numerical simulation under chemical erosion. (**a**) Uneroded specimen. (**b**) Erosion of 30-day specimen. (**c**) Erosion of 60-day specimen. (**d**) Erosion of 90-day specimen.

**Figure 12 materials-17-00701-f012:**
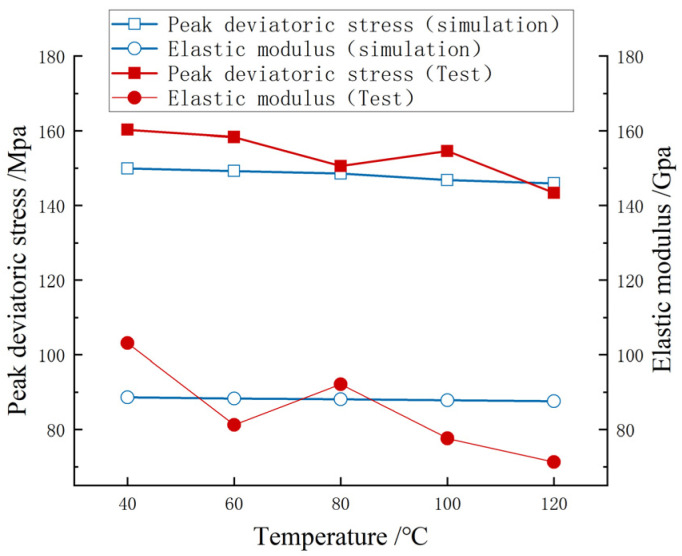
Verification and comparison of deterioration effect of carbonaceous shale under high temperature.

**Figure 13 materials-17-00701-f013:**
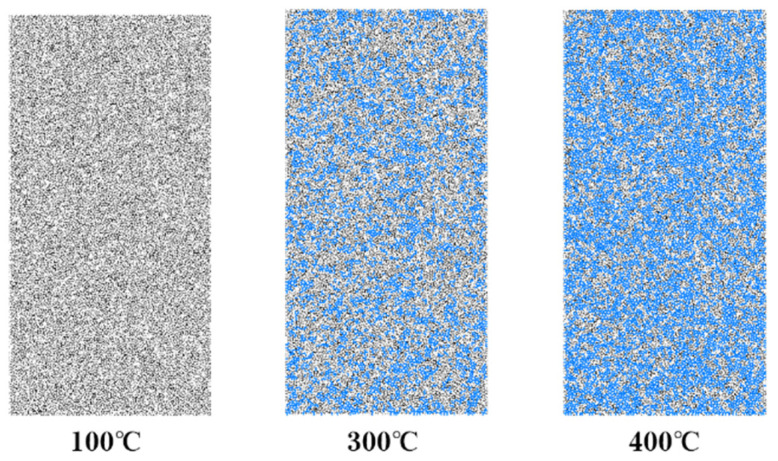
The internal microcrack variation of the specimen under different temperatures.

**Figure 14 materials-17-00701-f014:**
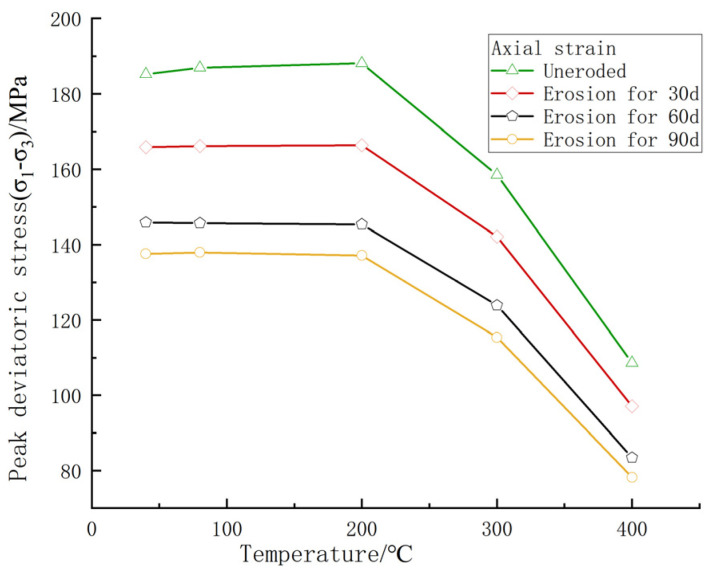
The effect of chemical erosion and high-temperature coupling on compressive strength.

**Table 1 materials-17-00701-t001:** Chemical solution preparation.

Solution Composition	Concentration/mol·L^−1^
Distilled water	—
10% dilute hydrochloric acid	—
Na_2_SO_4_ solution	0.1
NaCl solution	0.2

**Table 2 materials-17-00701-t002:** Mineral composition change under chemical weathering.

Erosion Duration	Quartz	Muscovite	Sodium Feldspar	Graphite
Uneroded	70%	16%	11%	3%
Erosion for 30 days	74%	13%	9%	4%
Erosion for 60 days	76%	12%	7%	6%
Erosion for 90 days	77%	11%	6%	6%

**Table 3 materials-17-00701-t003:** Microscopic parameters of carbonaceous slate.

Parameter	Numerical
Minimum particle diameter/m	5 × 10^−4^
Maximum particle diameter/m	12 × 10^−4^
Particle modulus/Gpa	34.5
Stiffness ratio	2.7
Particle tensile strength/Mpa	35
Particle cohesion/Mpa	50
Particle friction coefficient	0.5
Particle number	12,003

**Table 4 materials-17-00701-t004:** Mineral thermodynamic parameters.

Mineral	Coefficient of Thermal Expansion/10^−6^·°C^−1^	Specific Heat Capacity/C·(J·kg·°C^−1^) ^−1^
Quartz	24.3	1015
Muscovite	15.4	
Sodium feldspar	4.5	
Graphite	1.2	

**Table 5 materials-17-00701-t005:** Microcrack number inside the specimens under the coupled action of chemical erosion and high temperature.

Temperature/°C	Uneroded	Erosion for 30 Days	Erosion for 60 Days	Erosion for 90 Days
100	0	0	1	0
200	547	621	653	653
300	5190	5512	5608	5609
400	9930	10,229	10,342	10,333

## Data Availability

Data are contained within the article.

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
