# Peer review of "Study on the Degradation Effect of Carbonaceous Shale under the Coupling Effect of Chemical Erosion and High Temperature"

_materials, 2024, doi:10.3390/ma17030701_

Round 1
Reviewer 1 Report
Comments and Suggestions for Authors
In this study, scanning electron microscopy, XRD diffraction, as well as uniaxial and triaxial compression tests were conducted on carbonaceous shales with different erosion durations. Also, a numerical simulation method was developed based on the parallel bond model that considers the microstructure and mineral changes of chemical erosion at high temperature. The model was validated by comparison with experimental measurements. Mesoscopic damage and the effect of decay of carbonaceous shales under coupled erosion and high temperature were studied.
There is no clearly distinguished scientific contribution in the paper. Therefore, I suggest that the authors do a major revision.
Some minor remarks:
Figure and table captions should contain complete information about what is in the figures or tables (regardless of what is in the text of the paper).
Be sure to enter the abbreviations when logging in for the first time, and only once. For example, you introduced SEM twice. You introduce XRD when you use the expression four times.
Abbreviations FLAC3D, ABAQUS. You didn't introduce UMAT, PFC at all, and you don't use it in the text, so we wonder what the abbreviation will be.
Figure 6 under a) and b) should be an extension of the text of Fig. 6.
The resolution of the images is very poor and it is not clear what is on them. The font and font size should be in the font size of the text
Make a discussion of the entire investigation. Highlight the advantages and disadvantages of each method and calculation you made
On page 14...When the simulated temperature reached 200°C, the compressive strength loss of
417 the uneroded specimens, the specimens eroded for 30 days, 60 days, and 90 days were
418 -1.58%, -0.27%, 0.33%, and 0.39%, respectively. Not exactly how the four values relate to the three time periods? How can there be a negative value?
You did not specify the exact name of the instrumentation, city, country and year of manufacture
Supplement your reference list with relevant research you can compare with, so that your references are ~30%

Minor editing of English language required
Reviewer 2 Report
Comments and Suggestions for Authors
The work presented by Xiong et al. and entitled: "Study on the degradation effect of carbonaceous shale under the coupling effect of chemical erosion and high-temperature" is interesting, novel, original and is of great importance to the field of materials and tunnel Engineering. This work is suitable for publication in Materials, MDPI after considering the following comments:
(1) Why didn't the authors do SEM-EDX mapping to map the elements ? what about inductively coupled plasma (ICP) for quantitative elemental analysis ?
(2) Please rewrite the abstract. It is abit vague. The novelty and originality of the work needs to be highlighted for the readers of the journal.
(3) Page 2, Line 53: Please cite the following work:
https://doi.org/10.1080/10916466.2021.2017457
(4) Conclusion section needs to be improved and the English language revised.
Comments on the Quality of English LanguageThis paper is of good quality and suitable for publication in Materials, MDPI after minor revision.
Reviewer 3 Report
Comments and Suggestions for Authors
Revision of the manuscript “Study on the degradation effect of carbonaceous shale under the coupling effect of chemical erosion and high-temperature”
The subject matter covered in the article is significant. While the paper is generally well-written, there are some issues with English language expression and formatting.
The primary deficiency lies in the absence of an explanation from the authors regarding why the "typical surrounding rocks in southwestern China”, frequently experience deterioration effects due to the combined impact of chemical erosion and high temperatures (up to 300°C) in tunnels, thereby impacting the enduring stability of tunnel engineering?
Please explain better your motivation.
And there are issues with the format
Throughout the text: Citations must be included In the sentence and not at the beginning of the following sentence
For example:
Xie et al. [18]carried
Should be:
Xie et al. [18] carried
Line 44: … science. [1, 2]The … should be … science [1, 2]. The ….
Line 147: were used.[3] should be: were used [3].
Line 150: respectively. [24, 25]The should be respectively [24, 25]. The
Line 95:
Citations must be separated from the following word
[20]used should be [20] used
Line 95 and line 98:
PH values should be pH values
Line 197:
The size of Fig.6 must be bigger
Line 204 – 205:
The changes in triaxial peak strength at different time points in each group are shown in Figure 7.
Should be:
The changes in triaxial peak strength at different time steps in each group are shown in Figure 7.
Line 271 – 275:
Due to the safety, observation means, economic cost and other factors affecting the related laboratory tests of carbonaceous shale, the upper limit of temperature under the coupled action is low, and it is difficult to explore the deterioration effect under high temperatures above 150°C, and the mesoscopic damage of carbonaceous shale under high temperature is difficult to observe directly.
The sentence is difficult to understand (too large), please rewrite it!
Line 281 - 282
The linear parallel bond model with zero gap and failed damper corresponds to the parallel bond model.
The meaning is not clear, please explain it better
Fig.10
Must be bigger, text is difficult to read.
5. Conclusion
Please discuss the broader implications of the article's findings or arguments. Consider how the information presented contributes to the field or addresses the larger context. What are the potential implications for further research or practical applications?
Comments on the Quality of English Language
There are issues with the English (difficult to read) and the format
Throughout the text: Citations must be included In the sentence and not at the beginning of the following sentence
For example:
Xie et al. [18]carried
Should be:
Xie et al. [18] carried
Line 44: … science. [1, 2]The … should be … science [1, 2]. The ….
Line 147 : were used.[3] should be: were used [3].
Line 150 : respectively. [24, 25]The should be respectively [24, 25]. The
Line 95:
Citations must be separated from the following word
[20]used should be [20] used
Line 204 – 205:
The changes in triaxial peak strength at different time points in each group are shown in Figure 7.
Should be:
The changes in triaxial peak strength at different time steps in each group are shown in Figure 7.
Line 271 – 275:
Due to the safety, observation means, economic cost and other factors affecting the related laboratory tests of carbonaceous shale, the upper limit of temperature under the coupled action is low, and it is difficult to explore the deterioration effect under high temperatures above 150°C, and the mesoscopic damage of carbonaceous shale under high temperature is difficult to observe directly.
The sentence is difficult to understand (too large), please rewrite it!
